# OpenReview forum: "The Sword of DamocleSpeech: Demystifying Jailbreaking Attack in Discrete Token-based Speech Large Language Models"
_ICLR.cc/2026/Conference — ICLR 2026 Conference Withdrawn Submission_

### Official Review · Reviewer_XGRo · 2025-10-29

**Soundness:** 3
**Presentation:** 2
**Contribution:** 2
**Rating:** 2
**Confidence:** 3

**Summary:**

This paper investigates audio-based LLM jailbreaking, focusing on simple prefilling attacks for prompting models to generate unsafe responses. The authors evaluate both jailbreak success and the quality of generated responses, and observe that extending the prefilling token length does not consistently improve jailbreak effectiveness.

**Strengths:**

1.	Focusing on audio jailbreaking is an important and timely topic.

2.	The experiments cover a wide range of audio-capable LLMs, and the evaluation of pre-filling attacks is fairly comprehensive.

**Weaknesses:**

1.	Prefilling is already a well-studied jailbreaking technique in text-based LLMs, so applying it to audio LLMs is not particularly surprising in terms of effectiveness, and the method itself does not leverage any audio-specific characteristics.

2.	Evaluating jailbreak responses from multiple quality dimensions is reasonable and necessary, but this practice is already common in recent work, so it is difficult to claim novelty from the evaluation setup alone.

3. From Table 1, although the prefilling attack shows some effect, the success rate is still far from practical. It would likely be more meaningful to explore jailbreak methods that exploit audio-specific characteristics to achieve stronger and more realistic attack performance.

**Questions:**

NA

---

### Official Review · Reviewer_tUoM · 2025-10-30

**Soundness:** 2
**Presentation:** 1
**Contribution:** 2
**Rating:** 2
**Confidence:** 3

**Summary:**

This paper investigates prefilling jailbreak attacks on Speech Large Language Models. The authors demonstrate that inserting a single affirmative token (e.g., "Sure") can trigger simultaneous jailbreaks in both text and speech modalities across eight mainstream open-source SpeechLLMs. Testing on AdvBench, Hex-PHI, and HarmBench datasets, they find dramatic increases in attack success rates (e.g., Qwen2.5-Omni: 1.35% → 88.46%). The paper proposes a manifold-based interpretation of this phenomenon and introduces three evaluation metrics (usefulness, harmfulness, completeness) to assess jailbreak quality beyond binary classification.

**Strengths:**

I think the core prefilling finding is significant. That single-token prefilling can achieve high ASR across multiple SpeechLLMs is an important vulnerability. The jump from 1.35% to 88.46% for Qwen2.5-Omni is genuinely striking. Further, the paper covers a reasonably broad range of models (8) across three well-established benchmarks. And it seems reasonable to try and find a more trustworthy method for grading jailbreak success.

I think the paper scores highest for me on significance, since the jump in asr is genuinely quite surprising and worth paying attention to. It's not as strong on the other dimensions, as I'll touch on below.

**Weaknesses:**

The most notable issue with this paper, even if it's not the most negatively impactful aspect, is that it's presented really rather poorly, across writing and figures. The writing itself is frequently overly complex (I'm not sure "a multi-perspective assessment of safety bypasses" is the best example here but there are many such), and sometimes strangely metaphorical ("Once the 'valve' is opened" comes to mind when describing how prefilling opens up the model to generate harmful text). This alone would make the paper irritating to get through, but ultimately fine. However, what really tips this over for me is the lack of substantive captions to every figure, and how arcane the figures themselves are. I strongly advise the authors to rework their figures and provide adequate captions explaining their method - without which I find this paper, especially the manifold in fig 2, rather cryptic.

Further, I have serious concerns about the relevance of the title and framing of this paper. The central Sword of Damocles metaphor is invoked without explanation, mentioned only in the title. The title also suggests that the paper aims to "demistif[y] Jailbreaking Attack [sic] in Discrete Token-based Speech Large Language Models", but as far as I can tell, the most useful contribution in the paper is a particular (and interesting!) prefilling jailbreak result, which I'd have much preferred be a more substantive focus of the paper's framing.

On the methodology: If the paper is to, as the abstract claims, "reveal modal misalignment between discretized audio representations and textual embeddings", I'd need to see much more extensive analysis of the prefilling method and their metrics. Starting with the metrics, I overall applaud the authors for seeking a more rigorous and systematic method for judging jailbreak success. However, the authors do not provide sufficient justification for their suggested replacement. The authors tell us that "These three metrics [usefulness/harmfulness/completeness] are derived from ASR-based evaluation, exhibiting both interdependence and distinctiveness", but do not tell me why these are relevantly interdependent and/or distinct. (Incidentally, I'm confused whether these two suggestedly key properties are not entirely opposed to each other?). I'm also not sure that they're drawing a fair parallel between their judgment method and other LLM judges, or why it matters that their judgements are based on three binary classifications (the three axes above), instead of having a judge make an overall binary classification when passed these three factors in its rubric.

Other uncertainties I have, about the prefilling method itself, include questions such as how the affirmative tokens from table 4 were selected, why the authors think different affirmative tokens yield similar results, etc.

More minor, but throughout I get the sense that the literature cited is incomplete. One particularly salient example is that the only citation for RLHF (https://arxiv.org/pdf/2307.07176) looks to me to be not especially core to the RLHF literature - and is about RL with world models, which I don't think is especially relevant to the safety problems addressed in this paper under review. A more canonical RLHF citation would be appropriate, like Christiano et al., 2017; Stiennon et al., 2020; etc).

**Questions:**

1. Can you clarify what the manifold analysis actually demonstrates beyond the empirical observation that prefilling works? What specific insights does the mathematical framework provide that aren't evident from the empirical results alone?

2. Why does the paper title reference "Sword of Damocles" and what is being "demystified"? Please provide explicit justification for this framing.

3.How were the candidate affirmative tokens selected (Table 4)? Did you perform any systematic search, or were these chosen based on prior work?

4. Have you validated your three-dimensional evaluation framework (usefulness/harmfulness/completeness) against human judgments? How do you know these dimensions capture the most important aspects of jailbreak quality?

5. Table 2 shows that increasing prefilled tokens from 1 to 7 doesn't substantially improve usefulness, harmfulness, or completeness beyond the first token. Doesn't this contradict the emphasis on "token length" as a key parameter?

6. Figure 3 shows KL divergence over time steps, but I cannot interpret what the oscillating patterns mean. Can you explain what we should conclude from these plots, perhaps with commentary in the paper, or via a caption?

7. Can you provide confidence intervals or variance estimates for your main results (Table 1)?
The defense evaluation (Table 3) tests only one model with one defense. This seems insufficient to make general claims about defense effectiveness. Can you expand this evaluation?

---

### Official Review · Reviewer_1CAL · 2025-10-31

**Soundness:** 3
**Presentation:** 2
**Contribution:** 2
**Rating:** 2
**Confidence:** 3

**Summary:**

In this paper, the authors investigate the vulnerability of safety alignment of SpeechLLMs. They explore prefilling attacks and demonstrate that prefilling a single token is sufficient to subvert model safety across both speech and text modalities. In an attempt to understand this phenomenon, the authors conduct manifold analysis and demonstrate misalignment between discretized audio representations and textual embeddings.

**Strengths:**

- **Timely  topic**: Jailbreaking in SpeechLLMs is largely underexplored, making this investigation relevant for the community.

- **Comprehensive empirical evaluation**: The authors demonstrate results across several harmfulness benchmarks and evaluate an impresive suit of open-source SpeechLLMs.

- **Analysis**: The authors investigate potential underlying reasons via manifold analysis and explore potential defenses such as self-reminder mechanisms.

**Weaknesses:**

-  **Limited novelty**: The paper explores the same attack surface and vulnerability that has been extensively studied in existing jailbreaking papers on text-only LLMs (which the authors cite and refer to). While the application to SpeechLLMs is relevant, I struggle to grant this work sufficient novelty, as it does not meaningfully leverage the added audio modality. The attack essentially follows the standard prefilling approach from text-only LLMs. It is unclear what additional insights the audio modality provides. I would appreciate at least a comparative analysis showing how adding the audio modality affects vulnerability compared to text-only counterparts, or demonstrations of audio-specific attack vectors.

- **Overly permissive threat model**: The paper investigates a threat model where an attacker has access to model weights and the inference framework, which they tamper with by prefilling the first token. I struggle to justify the merit of exploring this threat model, as there appears to be no realistic deployment scenario where an attacker could prefill the model output in this manner. The practical implications and insights from this specific threat model remain unclear.

- The authors introduce a new harmfulness judge that takes into account harmfulness, usefulness, and completeness of the output. However, this judge is not supported by any human study demonstrating agreement, and it seems superfluous given the plethora [2] of existing jailbreaking judges some explicitly designed to judge usefulness, such as StrongREJECT [1].

[1] A StrongREJECT for Empty Jailbreaks
[2] https://github.com/llm-qc/judgezoo

**Questions:**

- Could you provide a comparison showing how the audio modality specifically affects vulnerability compared to text-only models?

- Line 287: The authors write "Wei et al. (2023b) introduced the concept of a 'jailbreak tax'..." Should this instead cite [1], the actual jailbreak tax paper?

- Line 363: There appears to be a typo in the LaTeX notation.

- Line 365: Extra period.

[1] The Jailbreak Tax: How Useful are Your Jailbreak Outputs?

---

### Official Review · Reviewer_UvE4 · 2025-11-03

**Soundness:** 2
**Presentation:** 2
**Contribution:** 2
**Rating:** 4
**Confidence:** 4

**Summary:**

This paper studied the prefilling attack to jailbreak the SpeechLLM. Specifically, the authors tried appending single affirmative token such as "Sure" before the malicious audio query, and observed that the success rate on some of the SpeechLLM such as Qwen2.5Omni can jump to 80%+.

**Strengths:**

This paper presents a systematic research on jailbreak attacks and safety alignment in SpeechLLMs and Omni models. It is interesting to see that only single affirmative prefilling token can jailbreak most of the mainstream SpeechLLM models.

**Weaknesses:**

I mainly have concerns regarding technical clarity.
- how is the success rate measured on SpeechLLM? I understand in text domain, we can use LLM as judge to measure if the text response from the LLM is harmful or not. However, in SpeechLLM, is that the case the generated speech response always match the text response even in the adversarial attack case?
- does the order of the adversarial token and audio query matter to the success rate? in this paper, the adversarial token is put before the audio query. Is it before system prompt or after? what if we put it after the audio query?
- did the author test the success rate of prefilling attack on the base LLMs from the SpeechLLM? For example, what is the success rate of prefilling attack on the Qwen2.5-7B which is the base LLM for Qwen2.5-Omni?

**Questions:**

please refer to the weakness section.

---

### Note · Authors · 2025-12-06

**Comment:**

We would like to express our sincere gratitude to the reviewers for their careful review and insightful comments. We have decided to withdraw this submission.

**Withdrawal Confirmation:**

I have read and agree with the venue's withdrawal policy on behalf of myself and my co-authors.